# Enhancing Classification Performance of fNIRS-BCI by Identifying Cortically Active Channels Using the z-Score Method

**DOI:** 10.3390/s20236995

**Published:** 2020-12-07

**Authors:** Hammad Nazeer, Noman Naseer, Aakif Mehboob, Muhammad Jawad Khan, Rayyan Azam Khan, Umar Shahbaz Khan, Yasar Ayaz

**Affiliations:** 1Department of Mechatronics Engineering, Air University, Islamabad 44000, Pakistan; hammad@mail.au.edu.pk; 2School of Mechanical and Manufacturing Engineering, National University of Science and Technology, Islamabad 44000, Pakistan; aakifmehboob@live.com (A.M.); jawad.khan@smme.nust.edu.pk (M.J.K.); yasar@smme.nust.edu.pk (Y.A.); 3National Centre of Artificial Intelligence (NCAI), Islamabad 44000, Pakistan; 4Department of Mechanical Engineering, University of Saskatchewan, Saskatoon, SK S7N5A9, Canada; rayyan.khan@usask.ca; 5Department of Mechatronics Engineering, National University of Sciences and Technology, H-12, Islamabad 44000, Pakistan; u.shahbaz@ceme.nust.edu.pk; 6National Centre of Robotics and Automation (NCRA), Rawalpindi 46000, Pakistan

**Keywords:** functional near-infrared spectroscopy, brain–computer interface, z-score method, channel selection, region of interest, channel of interest

## Abstract

A state-of-the-art brain–computer interface (BCI) system includes brain signal acquisition, noise removal, channel selection, feature extraction, classification, and an application interface. In functional near-infrared spectroscopy-based BCI (fNIRS-BCI) channel selection may enhance classification performance by identifying suitable brain regions that contain brain activity. In this study, the z-score method for channel selection is proposed to improve fNIRS-BCI performance. The proposed method uses cross-correlation to match the similarity between desired and recorded brain activity signals, followed by forming a vector of each channel’s correlation coefficients’ maximum values. After that, the z-score is calculated for each value of that vector. A channel is selected based on a positive z-score value. The proposed method is applied to an open-access dataset containing mental arithmetic (MA) and motor imagery (MI) tasks for twenty-nine subjects. The proposed method is compared with the conventional *t*-value method and with no channel selected, i.e., using all channels. The z-score method yielded significantly improved (*p* < 0.0167) classification accuracies of 87.2 ± 7.0%, 88.4 ± 6.2%, and 88.1 ± 6.9% for left motor imagery (LMI) vs. rest, right motor imagery (RMI) vs. rest, and mental arithmetic (MA) vs. rest, respectively. The proposed method is also validated on an open-access database of 17 subjects, containing right-hand finger tapping (RFT), left-hand finger tapping (LFT), and dominant side foot tapping (FT) tasks.The study shows an enhanced performance of the z-score method over the *t-*value method as an advancement in efforts to improve state-of-the-art fNIRS-BCI systems’ performance.

## 1. Introduction

Functional near-infrared spectroscopy (fNIRS) is a noninvasive optical imaging technique used to measure blood oxygenation changes as brain activity to develop a brain–computer interface (BCI) [1]. Among other noninvasive modalities used for BCIs like functional magnetic resonance imaging (fMRI) and electroencephalography (EEG); applications of fNIRS are increasing steadily in the BCI community [2,3,4,5]. fNIRS is a cheap, portable, and safe optical brain imaging technique used in state-of-the-art BCI systems to control and drive external devices using brain signals [1,6]. fNIRS is also used to analyse the brain at work and during complex everyday life situations [7,8,9,10]. fNIRS records brain activity as cortical blood oxygenation changes using two or more wavelengths of near-infrared lights ranging from 700 to 1000 nm. It measures the changes in oxy- and deoxy-hemoglobin (Δ*HbO* and Δ*HbR*) using the modified Beer–Lambert law [11,12]. The theoretical principles, advancements, and practical fNIRS applications have previously been described in detail [2,3,4,13]. fNIRS systems are portable, wearable, and overall user-friendly, making fNIRS a suitable choice for BCI. Recently, fNIRS has demonstrated successful and promising results in several BCI applications [14,15,16,17,18] and clinical applications [19,20] for brain imaging and brain signal acquisition purpose.

BCI systems have enhanced patients’ quality of life in clinics, hospitals, daily life activities, and at work [21,22]. A state-of-the-art BCI system includes recording brain signal, noise reduction, channel selection, extracting features, classification, and an application interface [2]. Initially, a suitable brain imaging modality is used to record brain signals. In the second step, preprocessing is performed which consists of detrending, removing physiological and instrumental noises, and cortical activity-based channel selection. Different methods can be used to select brain activation channels. These methods include the *t*-value method, baseline correction method, source analysis of brain activation, and others. In the third step, appropriate features are extracted, followed by brain signals classification using suitable machine learning algorithms. Finally, the control unit generates control commands using discriminated brain signals to control external devices. Brain signals with high signal-to-noise ratio, brain-activation-based channel selection, and suitable machine learning algorithms are essential components of a state-of-the-art BCI system. Recent studies have been conducted to enhance fNIRS-BCI systems’ performance by enhancing classification accuracy using different methods and techniques at every stage of the BCI system [2,14,16,23]. The studies showed efforts to improve classification accuracy by applying cortical-activity-based channel selection techniques, extracting novel features, determining optimal features, and optimal feature-combinations for fNIRS-BCI [24,25,26].

In BCI, selecting channels of interest (COI) or a region of interest (ROI) has manifold objectives: reducing processing time, reducing dimensionality, enhancing performance, and suitable brain region identification containing low noise signals. The selection of appropriate channels in EEG-based BCI has shown encouraging results [27]—Mainly filtering, wrapper, embedded, hybrid, and human-based techniques have been used for the purpose [27]. The sequential floating forward selection (SFFS) algorithm [28] and iterative relief based on distance from centre (IterRelCen) algorithm [29] were applied for channel selection of motor imagery (MI) tasks for EEG-BCI. Likewise, Pearson’s correlation method was used for channel selection of three different EEG datasets of MI tasks for EEG-BCI [3]. Li et al. [30] implemented three different strategies for channel selection for stroke patients. Feng et al. [5] used the CSP-rank channel selection methods along with multiband signal decomposition filtering for selection of optimal channels. Similarly, an attention-based convolutional recurrent neural network (ACRNN) was used to extract more discriminative features from EEG signals and improve the accuracy of emotion recognition [31]. Jin et al. [32] applied the bi-spectrum-based channel selection algorithm on MI tasks for EEG-BCI. For EEG-based BCI, cross-correlation [33], probabilistic mapping methods [34], contrast-to-noise-ratio [35], and principal component analysis [36] have also been used for channel selection and ROI identification.

However, for fNIRS-BCI, very few channel selection methods and techniques have been found in the literature, which includes the *t-*value method [37,38,39], baseline correction method [40], and hardware-based approach, i.e., bundled-optode method [41]. The *t*-value method is used excessively by researchers for selecting ROI and COI. This statistical-based approach considers only those channels that give a positive *t*-value (*t* > 0) or greater than the critical value (*t* > *t*_crt_) and *p* < 0.05, where the value of *t*_crt_ depends upon the degree of freedom (i.e., number of samples in the signal). The method includes a step-wise procedure of (a) generating a canonical hemodynamic response function (cHRF) using 2-gamma functions [42] or 3-gamma functions [43], (b) convolving cHRF with known stimulation interval (boxcar function) to get a modelled/desired hemodynamic response function (dHRF), (c) applying iteratively reweighted least squares algorithm to estimate parameters by using a general linear regression model with dHRF, and (d) final significance of the hypothesis is calculated through these estimated parameters. If the estimated parameters are positive, then specific stimulation is assumed active and vice versa. Sontosa et al. [44] described the method in detail for finding out the most significant stimulation through *t*-values. However baseline correction technique simply compares peak value of tasks with peak value of rest in brain signals. If peak value of task is greater than the peak value of rest, the channel is selected. In this paper, we propose a novel method, the z-score method that uses cross-correlation and z-scores for ROI/COI selection to enhance the fNIRS-BCI system’s performance.

In the proposed methodology, conventional steps of data acquisition and reduction of noise are followed. In the second step, brain-activation-based channel selection is performed. cHRF is calculated using two-gamma functions, followed by dHRF estimation. Cross-correlation is applied to dHRF and each channel of averaged trial. The max value of correlation coefficients is selected for each channel and forms another vector of all channels’ max values. The z-score is calculated for the vector of max values. If the z-score is greater than zero, then the channel is selected (z-score > 0). After that, features are calculated, and classification is performed. The proposed methodology is applied to an open-access dataset of left motor imagery (LMI), right motor imagery (RMI), and mental arithmetic (MA) in this study. All channels and the *t*-value-method-selected channels are used for verification, following the same classification steps. Results show that the classification accuracy achieved using the z-score method is significantly higher (*p* < 0.0167; Bonferroni correction applied) than the *t*-value method and by using all channels. For validation of the proposed method, it is also applied on another open-access database of 17 subjects having RFT, LFT, and dominant side FT tasks. The results also show better performance of the z-score method on the conventional *t*-value method, baseline correction method, and by using all channels.

## 2. Materials and Methods

### 2.1. Subjects/Participants

An open-access dataset of fNIRS single-trial classification for LMI vs. rest, RMI vs. rest, and MA vs. rest is used in this study [45]. The dataset contains brain signals of twenty-nine healthy subjects with mean age of 28.5 ± 3.7 years. There were 14 males and 15 females and none of them had any mental, neurological, or visual disorder. The experimental paradigm was explained in detail to subjects before taking the written consent. The experiments were conducted following the latest Declaration of Helsinki. The Ethics Committee approved this study for the Institute of Psychology and Ergonomics, Technical University of Berlin (approval number: SH_01_20150330).

### 2.2. Experimental Paradigm/Protocol

In the literature, researchers used mental arithmetic, visual tasks, letter padding, word generation, object rotation, motor imagery, motor execution, and music imagery as brain activities for data acquisition for fNIRS-BCI [22,40,46,47,48,49]. In this study, motor imagery of left- and right-hand and mental arithmetic were selected as the brain activities.

The subjects were seated on a comfortable chair facing a screen. They were asked to control their body movements and stay still as much as possible during data acquisition. The experiment contained three sessions of LMI, RMI, and MA tasks. Each session started with an initial rest of 60 s to set up the baseline followed by 20 repetitions of the selected tasks with 60 s of final rest at the end of the session. Each task started with 2 s of the visual introduction of the task. Then the subject was asked to perform a task for 10 s followed by rest for a period of 15–17 s. A short beep (250 ms) was played at the start and end of each task. Task instructions were displayed on the screen. During the rest period, the subjects were asked to relax—further details can be found in [45]. The experimental paradigm is shown in Figure 1.

#### 2.2.1. Motor Imagery (MI)

For MI tasks, subjects were asked to perform kinaesthetic MI, i.e., to imagine the opening and closing of their hands as they were grabbing a ball. As all subjects were naive, visual instruction using a black arrow pointing left or right side was displayed on screen for 2 s. A short beep sound was played before the arrow disappeared, followed by a fixation cross during the task period. The subjects were told to imagine opening and closing of the hand at a self-paced frequency of 1 Hz. Again, a short beep sound was played with ‘STOP’ written and displayed on the screen to end the task period. The fixation cross was also displayed on the screen during the rest period. This pattern was repeated twenty times in a single session keeping a balanced count of 10 trials for each LMI and RMI.

#### 2.2.2. Mental Arithmetic (MA)

For the MA task, subjects were instructed to perform the initial subtraction of a one-digit number from a three-digit number, e.g., 384-8, by displaying it on the screen for 2 s. They were asked to memorize the numbers shown on screen for subtraction. The screen changed to a black fixation cross for the task period with a short beep sound. During the task period of 10 s, the subjects were instructed to subtract the one-digit number from the result of the previous subtraction repeatedly. Followed by a 15–17 s rest period, subjects were allowed to relax, and a black fixation cross was also displayed on the screen. Just like the MI paradigm, task periods were ended by playing a short beep sound, and “STOP” written and displayed on the screen. Likewise, the MI paradigm, initial, and final rest of the 60 s, was included in the MA paradigm to set up a baseline.

### 2.3. Experimental Setup/Optode Placement

Fourteen emitters and sixteen detectors were used to record fNIRS signals with separation of 3 cm [50,51], resulting in thirty-six physiological channels. Nine channels were placed at the frontal cortex around Fp1, Fp2, and Fpz. Twelve channels were positioned at the motor cortex around C3 and C4 respectively. And three channels were placed at the visual cortex around Oz. Optodes were arranged according to the 10–20 international system as shown in Figure 2.

### 2.4. Signal Acquisition

fNIRS data were measured by NIRScout (NIRx GmbH, Berlin, Germany). Additionally, an opaque cap was used over a stretchy fabric cap to block ambient light, and also firm contact was observed between the optodes and scalp. The sampling frequency was set to 12.5 Hz. The brain imaging system used two wavelengths, 760 and 850 nm. Following the literature [11], the modified Beer–Lambert law (MBLL) was applied to convert brain signals recorded into Δ*HbO* and Δ*HbR*.
(1)$$\left[\frac{\Delta HbO\left(t\right)}{\Delta HbR\left(t\right)}\right]=\frac{{\left[\begin{array}{cc} \varepsilon_{HbO}\left(\lambda_{1}\right) & \varepsilon_{HbR}\left(\lambda_{1}\right)\\ \varepsilon_{HbO}\left(\lambda_{2}\right) & \varepsilon_{HbR}\left(\lambda_{2}\right) \end{array}\right]}^{-1}\left[\frac{\Delta ¥\left(t,\lambda_{1}\right)}{\Delta ¥\left(t,\lambda_{2}\right)}\right]}{d\times l}$$
where $\varepsilon_{HbO}\left(\lambda \right)\;\mathrm{and}\;\varepsilon_{HbR}\left(\lambda \right)$ are extinction coefficients of ΔHbO and ΔHbR in $µM^{-1}{\mathrm{cm}}^{-1}$ respectively, d is the differential path-length factor in [mm], l is the distance between emitter and detector in [mm], and Δ¥(t) is the absorbance difference of light source wavelength of $\lambda_{i}$ (where i = 1,2).

### 2.5. Signal Processing

Various noises like instrumental, physiological, and experimental noises contained by acquired hemodynamic signals had to be removed before feature extraction and classification [49]. Following the instructions [45] about preprocessing, ΔHbO and ΔHbR data were band-pass filtered using a fourth-order Butterworth filter with a passband of 0.03–0.15 Hz to remove physiological noises. A Savitzky–Golay filter was applied for smoothing [2] in MATLAB^®^ 2019b (The MathWorks, Inc., Natick, MA, USA). The averaged ΔHbO signals of all trials for channels 10, 12, and 22 for the MA, LMI, and RMI tasks after noise removal are shown in Figure 3 for an example subject.

### 2.6. Channel Selection/Channel of Interest/Region of Interest

In conventional BCI systems, either all channels are used, or channels are selected based on brain activation. In this study, the z-score method for COI/ROI is proposed and used for channel selection based on brain activation. The researchers have used the *t*-value method excessively for this purpose; therefore, it is included in the study. The conventional and proposed methodology is shown in Figure 4.

#### 2.6.1. *t*-value Method

The *t*-value method is an estimation-based channel selection or COI or ROI approach in which channels with a positive *t*-value are selected. Alternatively, a threshold value of ‘*t*’ (*t* > *t*_crt_) can also be set for the selection of active channels. In that case, the degree of freedom (*k* − 1) is used to determine the threshold value, where ‘*k*’ is the number of samples in an activity. The *t*-value method determines cortical activation through statistical estimation by fitting the linear regression model [44]. The estimation can be calculated by fitting dHRF, with measured hemodynamic response function resulted from cortical activation. It can be formulated as given below:(2)$$h_{j}^{i}\left(k\right)=\phi_{j}^{i}h_{M}\left(k\right)+\psi_{j}^{i}.1+\varepsilon_{j}^{i}$$

The term on the left side of the equation i.e., $h_{j}^{i}\left(k\right)\in R^{k\times 1}$ is the measured response function in which ‘*k*’ is the number of samples in each stimulus, subscript ‘*j*’ denotes the stimulus number, and superscript ‘*i*’ represents the channel number. ‘ϕ’ is the unknown coefficient to be estimated, the coefficient ‘ψ’ is multiplied by column vectors of $1\in R^{k\times 1}$ for correction of baseline drift in the signal, and $\varepsilon \in R^{k\times 1}$ is the error term in the linear regression method. The unknown coefficients ‘ϕ’ are estimated through *robustfit* function in MATLAB^®^. $h_{M}\left(k\right)\in R^{k\times 1}$ is calculated by convolution of cHRF $h_{c}\left(k\right)$ with boxcar function s(k)
$h_{c}\left(k\right)$ and can be modeled using two-gamma functions [42,43,52], as shown below:(3)$$h_{c}\left(k\right)=A\left[\frac{k^{\alpha_{1}-1}\beta_{1}^{\alpha_{1}}e^{-\beta_{1}k}}{\Gamma \left(\alpha_{1}\right)}-c\frac{k^{\alpha_{2}-1}\beta_{2}^{\alpha_{2}}e^{-\beta_{2}k}}{\Gamma \left(\alpha_{2}\right)}\right]$$

The parameter ‘*A*’ sets the amplitude, ‘*α*_1_′ and ‘*α*_2_′ set the peak and undershoot delays, respectively. In contrast, ‘*β*_1_′ & ‘*β*_2_′ set dispersions of the peak and undershoot respectively, ‘*c*’ is the ratio of the peak to the undershoot, and ‘*Γ*’ is the gamma distribution. $h_{M}\left(k\right)$ can be calculated using the formula
(4)$$h_{M}\left(k\right)=\overset{k}{\underset{n=1}{\sum }}h_{c}\left(n\right)s\left(k-n\right)$$

And the boxcar function is:(5)$$s\left(k\right)=\{\begin{array}{c} 0,\;if\;k\in rest\\ 1,\;if\;k\in task \end{array}$$

The boxcar function is a unit step function having a value of ‘0’ for the rest period and ‘1’ for the task period. After estimating the coefficient ‘ϕ’, its statistical significance is calculated by the ratio of the estimated coefficient and its standard error (SE). The said statistical significance is also called as ‘*t*-value’. Its positive or threshold value greater than the critical value shows that the channel is active or otherwise.
(6)$$t_{j}^{i}=\frac{\phi_{j}^{i}}{SE\left(\phi_{j}^{i}\right)}$$

The above formula gives a *t*-value in *i*-th channel of *j*-th stimulus. In our case, active channels are considered which have *p*-value less than 0.05 and a *t*-value greater than ‘*t*_crt_’, which is 1.65 (degree of freedom is 299, i.e., *k* − 1). This method was initially used to measure the statistical significance of channel [44]. The step-by-step procedure of the *t*-value method is shown in Figure 5a.

#### 2.6.2. z-Score Method

The z-score method uses cross-correlation as the mutual relationship between two signals and measures the strength of the relationship among the acquired signal and dHRF. Cross-correlation matches two signals temporally to find out the strength of similarity between each other, and mathematical expression is given in the equation below
(7)$$r_{xy}\left(\tau \right)=\sum_{-\infty }^{\infty }x\left(t\right)y\left(t-\tau \right)$$
where τ is the time-lag between x(t) and y(t), and the value of $r_{xy}$ denotes the difference between acquired signal x(t) and modelled signal y(t). This cross-correlation has been used earlier for finding the relationship of the potential dominant channel with its adjacent channels by observing delay in response between the channels [5]. In the current study, the dHRF signal is swept over the measured signal, and the integral of its product is found at each discrete position ‘*t*’. The maximum value of the integral product, i.e., the correlation coefficient, is selected for each channel showing the temporal similarity between two signals at that time instant ‘τ’.
(8)$$r^{i}=max\left(\overset{k}{\underset{t=-k}{\sum }}h^{i}\left(t\right)h_{M}^{i}\left(t-\tau \right)\right)$$

The average trial value of measured response is taken for each stimulus type (i.e., LMI, RMI, and MA), and afterward, cross-correlation is calculated with dHRF. Maximum strength of similarity occur when τ is selected for task vs. task intervals of measured and desired hemodynamic response function to overlap, it is also the highest value of cross-correlation coefficient. If τ is selected for rest vs. rest or rest vs. task or task vs. rest intervals of measured and desired hemodynamic response function to overlap, it will give lower values of cross-correlation coefficient. And if τ is selected for complete intervals i.e., calculate cross-correlation coefficient for complete time period, the highest value remains the same, as shown in Figure 6a. The maximum value of correlation coefficient ‘r’ is selected for each channel ‘i’, between measured hemodynamic response function $h\left(t\right)\in R^{k\times 1}$ and dHRF $h_{M}\left(t\right)\in R^{k\times 1}$, where ‘k’ is the number of samples in the signal. The vector $r^{i}$ contains maximum values of cross-correlation coefficients for each channel. The magnitude of each maximum value varies for each channel and forms a new range as shown in Figure 6b. The z-score measures the distance of raw score from mean value i.e., how far from mean a data point is in population. In this study, z-score represents the channel activation in the form of matching and strength of similarity based on the value of the cross-correlation coefficient. A positive z-score represents higher strength and similarity and a negative z-score shows lower strength and no matching. The z-scores of the channel vary as the (max of) cross-correlation coefficient value varies with respect to task, as shown in Figure 6 for (c) LMI, (d) RMI, and (e) MA. Additionally, the z-score varies with subject. z-scores of vector containing maximum values of cross-correlation coefficients for each channel are then calculated using the formula.
(9)$$z^{i}=\frac{\left(r^{i}-\bar{r}\right)}{\sigma_{r}}$$
where ‘$\bar{r}$’ is the mean value of correlation coefficients and ‘$\sigma_{r}$’ is the standard deviation. Only those channels are selected which have a positive z-score (i.e., *z* > 0). The step-by-step procedure of channel selection using the z-score method is shown in Figure 5b.

Both the *t*-value method and z-score method are used to select cortical-activation-based channels.

### 2.7. Feature Extraction

#### 2.7.1. Statistical Features

In fNIRS-BCI, statistical measures such as peak, mean, variance, kurtosis, skewness, and slope are extracted as features for classification [2,24,53,54,55,56]. However, mean and peak and mean, peak, and slope were optimal two- and three-feature combinations to achieve enhanced classification accuracies for the fNIRS-BCI system [57]. In this study, the mean, peak, and slope are used as features for the fNIRS-BCI problem classification. All features are calculated for ΔHbO spatio-temporally. Spatio-temporal features are calculated using a two-step procedure: (1) averaging all channels (spatial average) and (2) aggregating using a statistic across each task window (temporal statistic).

Mean is calculated as:(10)$$Mean=\mu =\frac{1}{n}\overset{k}{\underset{x=1}{\sum }}B_{x}$$
where $B_{x}$ is the input signal such as ΔHbO(t) and n is the total number of observations, aggregated by mean function of MATLAB^®^.

The slope is aggregated using MATLAB^®^
polyfit function, which fits the line to all input data points.

The peak is the maximum value of the signal, calculated using max function of MATLAB^®^.

#### 2.7.2. Normalization

Features are normalized by rescaling using the following equation:(11)$$Y^{\prime }=\frac{Y-min\left(Y\right)}{max\left(Y\right)-min\left(Y\right)}$$
where $Y^{\prime }$ is the normalized feature, and Y is the original feature values. This normalization has been applied to all features before classification. The final feature-matrix calculated is of size 20 × 3 for each task.

### 2.8. Linear Discriminant Analysis (LDA)

The linear discriminant analysis draws a hyper-plane in feature space to discriminate between classes. The hyper-plane is drawn based on minimizing the inter-class variance and maximizing the distance between classes mean. The optimal projection matrix to maximize Fisher’s criterion is formulated as
(12)$$J\left(X\right)=\frac{X^{T}S_{B}X}{X^{T}S_{W}X}$$
where $S_{W}$ is with-in class scatter matrix and $S_{B}$ is a between-class scatter matrix, defined as:(13)$$S_{W}=\overset{n}{\underset{i=1}{\sum }}\underset{y_{k}\in class\;i}{\sum }\left(y-\mu_{i}\right){\left(y-\mu_{i}\right)}^{T}$$
(14)$$S_{B}=\overset{n}{\underset{i=1}{\sum }}k_{i}\left(\mu_{i}-\mu \right){\left(\mu_{i}-\mu \right)}^{T}$$

In the above equation $\mu_{i}$ denotes the sample mean of class *i*, and μ represents the total mean of all samples, n is the total number of classes, $k_{i}$ designates the number of samples of class *i*, and k is the total number of samples. The largest eigenvalues contained by optimal vector X is calculated by Equation (12) as a generalized eigenvalue problem. To estimate classification performance leave-one-out cross-validation (LOOCV) is used. The dataset is divided into training and testing sets, to ensure separation of data for training and testing of classifier for each channel selection method and activity used. Due to a limited number of samples i.e., twenty, LOOCV is applied. There is one sample for testing and nineteen for training the classifier, repeated twenty times. In MATLAB^®^ the following functions were used; *cvpartition* for data partition in folds, *classify* for classification, and *crossval* for cross-validation purposes.

## 3. Results

The analysis shows that both methods vary in channel selection and the total number of selected channels for a specific task. The *t*-value method measures the activation as statistical significance of signal of channel, while the z-score measures the activation as z-score of (max of) cross-correlation coefficients with respect to all channels population. The activation map is drawn for both methods using normalized *t*-values and z-scores. Figure 7 shows the cortical activation-map of the *t*-value and z-score method for MA, LMI and, RMI tasks. The figure shows the difference in measuring cortical activation by using both methods. It can be seen in Figure 7 that cortical activation does not occur in all channels for a specific activity, and also not in all channels of the designated region. For MA tasks the proposed method selected major channels from the prefrontal region (details can be found in Appendix A) and the visual cortex (instructions were displayed on screen), however some other channels from the motor cortex also showed a positive z-score value. Similarly for LMI and RMI major channels are selected from the motor and prefrontal cortex (details can be found in Appendix A) and the visual cortex (instructions were displayed on screen). The areas of activation found are similar to designated areas of neural activity [55]. The selection of extra channels may be due to human error, lack of concentration, multiple thinking during experiment, or induced neuronal activity. This also varies with respect to activity and subject. In Figure 7, the spatial difference of identifying cortical activation is found in both methods because of the fact that both methods apply different scientific and mathematical principles. The *t*-value method uses statistical significance by GLM and the z-score method applies signal matching through cross-correlation. A similar pattern of differences is found in all subjects. The proposed method is used to select channels having cortical activity during particular tasks. The number of chosen common channels between the *t*-value and z-score methods also varies for a specific task. For the MA task, the range of the total number of selected channels using the z-score method is 14–20 channels. The *t*-value method selected 6–36 channels, and the number of common channels between both methods varies from 2–20 channels. Similarly, for the LMI task, the chosen z-score method channels range from 14–20 channels, while the *t*-value has a range of 2–24 channels, and the number of common channels between both methods ranges from 1–20 channels. Likewise, for the RMI task, z-score-method-nominated channels range from 16–21 channels, whereas the *t*-value method selected 6–36 channels, and the amount of common channels ranges from 2–20. All three tasks of MA, LMI, and RMI were analyzed for all twenty-nine subjects. A plot of total number of channels selected by *t*-value, z-score method, and common channels for the MA, LMI, and RMI task is shown in Figure 8a–c respectively. Details of channels selected using the *t*-value and z-score method are available in Appendix A for the MA, LMI, and RMI tasks, respectively.

After selecting channels, Spatio-temporal features of the mean, peak, and slope were extracted from nominated channels’ data. In addition to the *t*-value method, all channel data were also used to compare the results. The feature scatters plot of the *t*-value-method-selected channels, the z-score-method-chosen channels, and all channels were drawn for each task. The feature scatter plot for MA, LMI, and RMI tasks for subject 28 is shown in Figure 9, Figure 10 and Figure 11, respectively.

The average classification accuracy obtained using channels selected by the z-score method is 87.2 ± 7.0%, 88.4 ± 6.2%, 88.1 ± 6.9% for LMI vs. rest, RMI vs. rest, and MA vs. rest; respectively. While the average classification accuracies yielded by using channels selected by the *t*-value method are 74.5 ± 9.3%, 70.3 ± 14.2%, and 73.9 ± 12.2% for LMI vs. rest, RMI vs. rest, and MA vs. rest; respectively. Likewise baseline-correction-technique-selected channels achieved classification accuracies of 79.3 ± 10.7%, 78.4 ± 13.3%, and 79.1 ± 18.1% for LMI vs. rest, RMI vs. rest, and MA vs. rest; respectively. However, all channels’ data achieved classification accuracies of 77.7 ± 8.9%, 75.0 ± 10.8%, and 77.5 ± 9.6% for LMI vs. rest, RMI vs. rest, and MA vs. rest; respectively. Table 1 shows the detailed subject-wise classification accuracies using the z-score method, *t*-value method, and all channels’ data for the MA, LMI, and RMI tasks. Figure 12 shows the bar chart for obtained accuracies using the z-score method, *t*-value method, and all channels for the MA, LMI, and RMI tasks. The better results obtained by the z-score method compared to conventional *t*-value and all channels’ data are statistically verified by applying a two-tailed paired sample Student’s *t*-test. For two comparisons, Bonferroni [58] correction was used to find the adjusted confidence interval level of 0.0167. Table 2 shows the *p*-values obtained for two comparisons for each task: the z-score method vs. *t*-value method and the z-score method vs. all channels. It can be seen that the z-score-method-selected channels’ performance is significantly better (*p*-value < 0. 0167) than the *t*-value method and all channels for MA, LMI, and RMI fNIRS-BCI.

## 4. Validation

The validation of proposed z-score method has also been performed on a publicly available fNIRS dataset. Left- and right-hand finger tapping and dominant foot tapping tasks were included in the dataset for 17 subjects [59]. Each of the aforementioned activities consists of 25 trial data. Six conventional spatio-temporal features that include signal mean, peak, slope, skewness, kurtosis, and variance of ΔHbO signal are extracted. LDA classifier is applied on the said extracted features for a two-class fNIRS-BCI problem.

The average classification accuracies yielded by using the selected channels through the proposed z-score-method-based selected channels are 72 ± 8%, 66 ± 9.8%, and 67.41 ± 9.5% for RFT, LFT, and FT tasks respectively. In comparison, the *t*-value-method-based selected channels resulted in average classification accuracies of 54.47 ± 10%, 54.12 ± 13.8%, and 56.24 ± 9.3% for RFT, LFT, and FT tasks respectively. Nonetheless, average classification accuracies obtained for RFT, LFT, and FT tasks by selecting all channels remained 54.71 ± 10.3%, 54.47 ± 14.2%, and 54.12 ± 11.1% respectively. However, the classification accuracies resulted using baseline-correction-selected channels for RFT, LFT, and FT tasks were 52.82 ± 7.4%, 51.06 ± 9.5%, and 55.53 ± 12.2% respectively. The results achieved using z-score method have significantly (*p*-value < 0.0167) better performance as compared to the results of the *t*-value method, the baseline correction technique and using all channels for two-class fNIRS-BCI problem.

## 5. Discussion

In the present study, the authors proposed a new method of selecting cortical-activation-based channels to increase fNIRS-BCI performance, especially in terms of classification accuracy and COI/ROI. In the literature, recent studies have also focussed on enhancing classification accuracies of fNIRS-BCI systems by optimal classification technique [60], optimal feature selection [24,54], optimal feature-combination [57], general linear model [25], vector-based phase analysis [26,61,62], *t*-value method [22,37,41,63], cross-correlation [33], and dominant channel selection [64]. Accurate and reliable fNIRS-BCI performance may lead to producing applications in neurorobotics, rehabilitation, clinical BCI for monitoring and analysis, and neuroergonomics [10,65,66,67].

In previous studies, bundled optode configurations have been used to precisely identify active regions of the brain [41] in spatially resolved spectroscopy. Santosa et al. [44] first applied the *t*-value method to select hemodynamic responses with positive *t*-values. Another study detected ROI against different sound stimuli by placing optodes on both right and left hemispheres [63]. It is worth mentioning here that different channels against different subjects and stimuli were obtained. The baseline correction method was used in hybrid-BCI to select active channels by calculating and comparing the maximum values during rest and corresponding task stages [40]. The cross-correlation method was used to identify potentially dominant channels in both hemispheres for pain-related cortical activations. First visual inspection was used to identify potential dominant channels followed by calculating the delay of response—The adjacent active channels were selected [33]. In the present study, a novel method for cortical-activity-based channel selection is proposed and validated for three different brain activation fNIRS-BCIs. The results have shown improved classification performance as compared to previous methods. The proposed method is compared to the conventional excessively used *t*-value method and without channel selection using all channels’ data. The proposed method uses cross-correlation to match and measure strength of similarity between dHRF and recorded brain signals. Followed by forming a vector of each channel correlation coefficients’ max values, and the z-score is aggregated for each value of that vector. On the basis of z-score value (z-score > 0) a channel is selected. This study shows the improvement in classification accuracies of three activity-based fNIRS-BCIs using the z-score method compared to the *t*-value method for channel selection and without channel selection, i.e., using all channels’ data. The classification accuracies are improved from 77.7 ± 8.9% to 87.2 ± 7.0% for MA vs. rest, 77.6 ± 9.6% to 88.1 ± 7.0% for LMI vs. rest, and 75.0 ± 10.9% to 88.4 ± 6.3% for RMI vs. rest. The channel selection results also verify that activation of channels is not uniform among different subjects due to variation in brain sizes. Similarly, a specific task is associated to a certain brain region—that is why identification of correct COI/ROI is extremely important.

The performance improvement may be since the z-score-method-selected channels represent brain activity more informatively and specifically than using the *t*-value method. The *t*-value method finds out the statistical significance by fitting the actual response to the estimated coefficients’ desired response. The *t*-value is the ratio of weighting coefficients to its standard error, and its positive (*t* > 0) or threshold value (*t* > *t*_crt_ and *p*-value < 0.05) decides whether the channel activity is significant or not. The t-value method was first used by Santosa et al. to measure the statistical significance of a channel [44]. Later studies used this method to select channels based on cortical activity in terms of a channel’s statistical significance. However, the z-score method measures the strength of similarity as cross-correlation coefficient between the desired response and measured response. The maximum value of the cross-correlation coefficient represents the extent of similarity and matching between desired and measured responses. Furthermore, the z-score value is the measure of distance from the mean value in a data, and a positive z-score value decides whether the raw score is at the right side of bell’s curve within a population as the channel’s cortical activation. The proposed method selects a channel from a designated region with respect to activity along with a few other channels. The extra channels are selected show a positive z-score because of human error, which includes lack of concentration towards the experiment, multiple thoughts during the experiment, and arbitrary neuronal activity. Cortical-activity-based channels possess intrinsic brain activation information, which plays an essential role in improving the fNIRS-BCI system’s classification accuracy. The z-score method can be used to identify cortical-activity-based brain activation regions, subregions, and channels to analyze, perform, and develop state-of-the-art fNIRS-BCI applications, including prosthetics, exoskeleton controls, and communications with stroke or locked-in syndrome patients.

This study has few limitations, including the fact that it applies to a single activity at a time as specific task is associated with a particular brain region. Furthermore, activation of subject-based specific channels occurs due to different brain sizes. The z-score method selects major chunks of channel from a designated region and a few other channels from other regions as well; this occurrence was also found in the *t*-value method. Further improvement can be made to reduce the selection of undesignated channels by only performing analysis on subregions. Moreover, the study includes only an LDA-based classification model because of its low computational cost and high-speed performance. LDA is a commonly used classifier for the fNIRS-BCI system [60]. Other machine learning algorithms may also be used for analysis and may perform better [55].

## 6. Conclusions

The aim of this study was to improve classification accuracy for a fNIRS-BCI system by selecting cortical-activity-based channels. The z-score method selects cortical-activity-based channels based on cross-correlation coefficients and z-score value (z > 0). Average classification accuracies achieved for MA vs. rest, LMI vs. rest, and RMI vs. rest by using the proposed method are significantly (*p* < 0.0167) higher than the *t*-value method and without channel selection, i.e., using all channels for classification. The results show enhanced performance for the proposed method over conventional methods as an advancement in efforts to identify cortically active channels/regions and to improve the classification performance of state-of-the-art fNIRS-BCI systems.

## Figures and Tables

**Figure 1 sensors-20-06995-f001:**
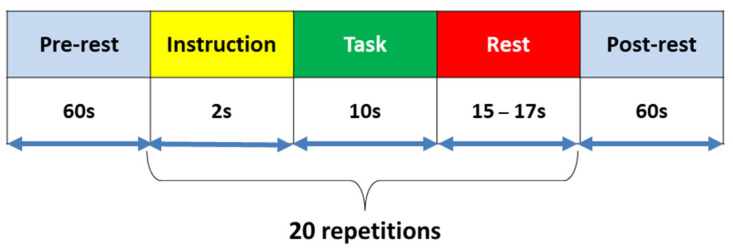
Experimental paradigm for data acquisition. After the initial 60-s rest, a single trial consisted of a 2-s visual instruction period, then 10 s left motor imagery (LMI), right motor imagery (RMI), motor imagery (MI), and mental arithmetic (MA) tasks followed by a 15–17 s rest.

**Figure 2 sensors-20-06995-f002:**
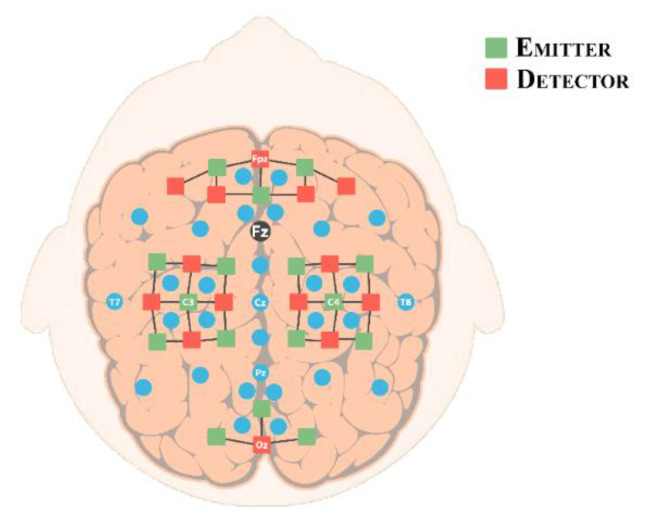
Optodes were placed at the frontal, motor, and visual cortex following the 10–20 international system [45]. Green and red squares represent emitters and detectors, respectively. Fourteen emitters and sixteen detectors were used to record functional near-infrared spectroscopy (fNIRS) signals with separation of 3 cm, resulting in a total of thirty-six.

**Figure 3 sensors-20-06995-f003:**
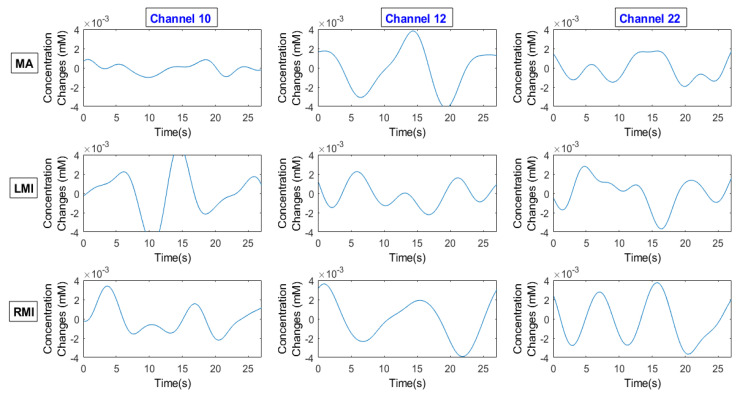
Averaged ΔHbO of all trials of channels 10, 12, and 22 for tasks and rest of the MA, LMI, and RMI.

**Figure 4 sensors-20-06995-f004:**
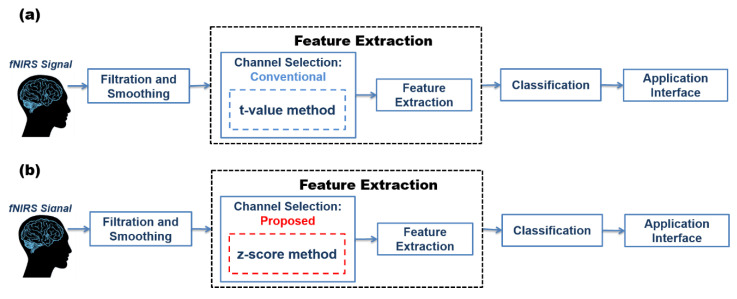
Methodology of (**a**) conventional and (**b**) proposed brain–computer interface (BCI) system.

**Figure 5 sensors-20-06995-f005:**
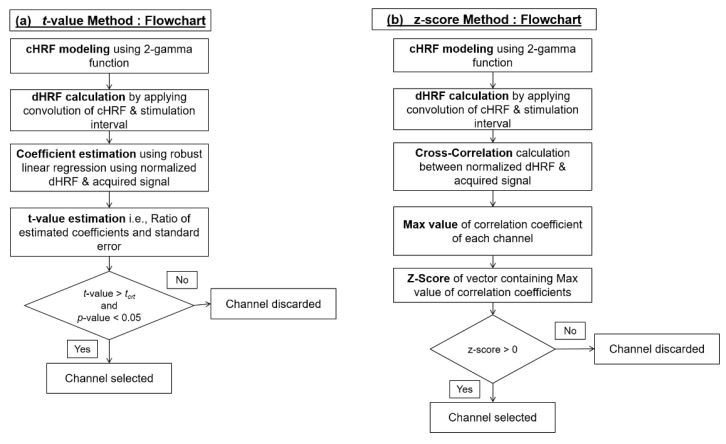
Step-by-step procedure of (**a**) *t*-value and (**b**) z-score method.

**Figure 6 sensors-20-06995-f006:**
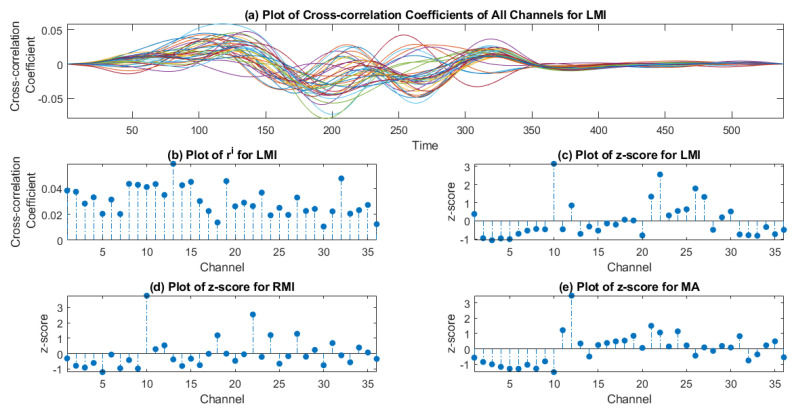
(**a**) Plot of cross-correlation coefficients of all channels for LMI for subject 28, (**b**) plot of $r^{i}$; vector containing maximum value of all channels for LMI for subject 28 and plot of z-score for (**c**) LMI, (**d**) RMI, and (**e**) MA task of subject 28.

**Figure 7 sensors-20-06995-f007:**
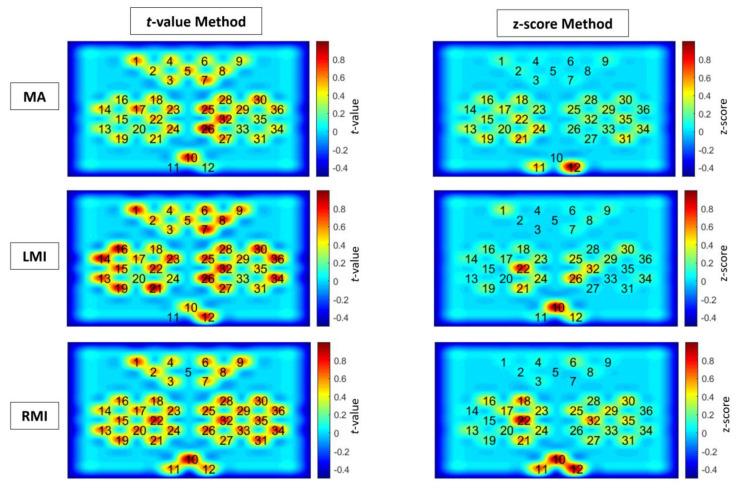
Activation map of *t*-value and z-score method for LMI, RMI, and MA tasks, for example subject 28.

**Figure 8 sensors-20-06995-f008:**
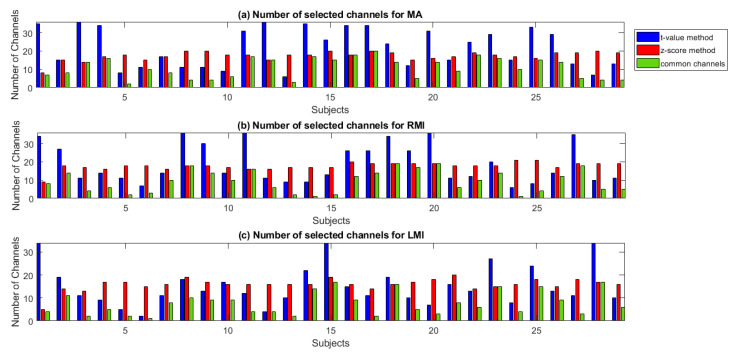
Total number of channels selected by *t*-value, z-score method, and common channels for (**a**) MA, (**b**) RMI, and (**c**) RMI task.

**Figure 9 sensors-20-06995-f009:**
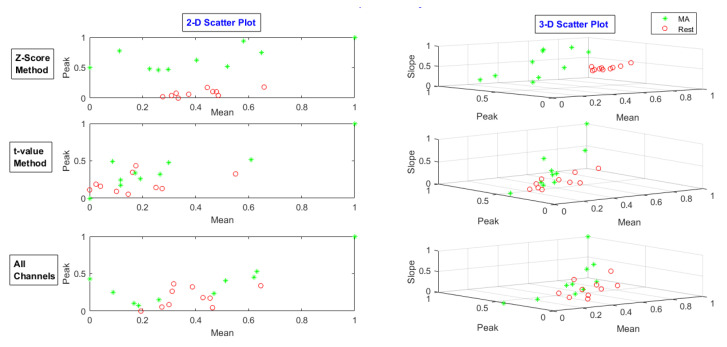
Feature scatter plot of the *t*-value-method-selected channels’ data, z-score-method-selected channels’ data, and all channels’ data for MA task.

**Figure 10 sensors-20-06995-f010:**
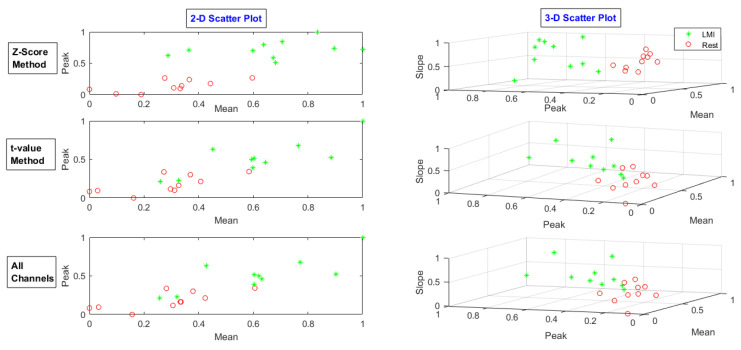
Feature scatter plot of the *t*-value-method-selected channels’ data, z-score-method-selected channels’ data, and all channels’ data for LMI task.

**Figure 11 sensors-20-06995-f011:**
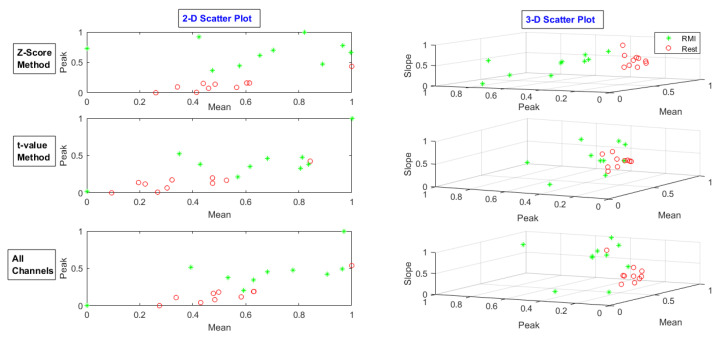
Feature scatter plot of the *t*-value-method-selected channels’ data, z-score-method-selected channels’ data, and all channels’ data for RMI task.

**Figure 12 sensors-20-06995-f012:**
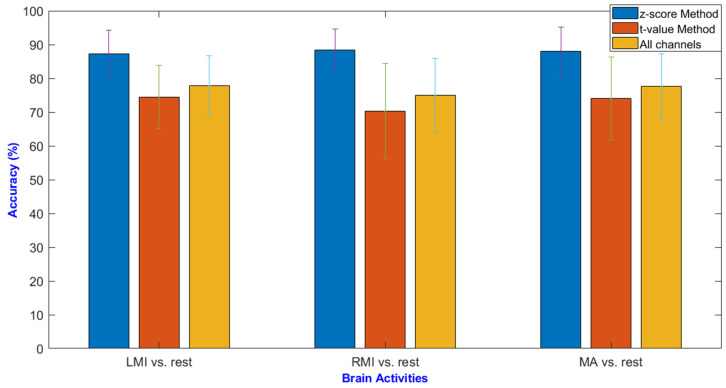
Average classification accuracies for the z-score method, *t*-value method, and all channels’ data for MA, LMI, and RMI tasks.

**Table 1 sensors-20-06995-t001:** Subject-wise classification accuracies by using the z-score method, *t*-value method, and all channels’ data for MA, LMI, and RMI tasks.

	MA			LMI			RMI		
	z-Score Method	*t*-Value Method	All Channels	z-Score Method	*t*-Value Method	All Channels	z-Score Method	*t*-Value Method	All Channels
	(%)								
Sub 1	75	85	90	90	70	55	90	70	75
Sub 2	90	70	80	85	80	90	85	65	75
Sub 3	80	70	70	95	75	65	90	80	75
Sub 4	95	80	80	85	70	75	85	80	65
Sub 5	90	65	55	90	55	80	85	75	65
Sub 6	90	45	55	95	80	85	90	60	65
Sub 7	90	70	80	80	65	65	90	55	85
Sub 8	90	60	90	90	80	85	85	85	85
Sub 9	90	80	60	75	65	70	100	90	85
Sub 10	85	80	85	80	70	75	85	80	70
Sub 11	95	90	85	85	70	70	95	70	70
Sub 12	90	75	75	95	80	80	75	65	65
Sub 13	95	45	70	90	70	70	90	90	80
Sub 14	90	70	70	95	85	90	95	65	65
Sub 15	95	70	75	85	85	85	90	75	90
Sub 16	75	75	70	90	80	75	90	60	80
Sub 17	85	80	80	95	80	85	95	80	85
Sub 18	85	90	90	90	70	85	90	90	90
Sub 19	85	80	85	70	85	85	80	85	80
Sub 20	95	95	85	95	50	80	95	60	60
Sub 21	75	80	80	85	75	80	85	60	90
Sub 22	85	75	85	80	60	70	95	80	75
Sub 23	80	75	80	95	80	80	80	65	70
Sub 24	95	60	80	80	70	85	90	35	75
Sub 25	100	90	80	95	85	90	75	45	50
Sub 26	90	85	90	80	80	70	95	70	75
Sub 27	80	65	75	85	75	65	95	85	85
Sub 28	100	60	70	95	85	85	90	75	90
Sub 29	85	80	80	80	85	80	80	45	55
Average	**88.1**	**74.0**	**77.6**	**87.2**	**74.5**	**77.8**	**88.4**	**70.3**	**75.0**

**Table 2 sensors-20-06995-t002:** Statistical significance of the z-score method.

	Bonferroni Correction Applied (*p* < 0. 0167)		
	LMI vs. Rest	RMI vs. Rest	MA vs. Rest
z-score method vs. *t*-value method	2.21 × 10^−7^	5.34 × 10^−8^	1.47 × 10^−6^
z-score method vs. all channels	3.50 × 10^−5^	3.54 × 10^−7^	1.37 × 10^−5^

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
