# Peer review of "Enhancing Classification Performance of fNIRS-BCI by Identifying Cortically Active Channels Using the z-Score Method"

_sensors, 2020, doi:10.3390/s20236995_

Round 1

Reviewer 1 Report

This paper proposed a channel selection method based on z-score to improve fNIRS-BCI performance. On the whole, the part of introduction is not enough, and more experiments should be added to demonstrate the advantage of the proposed method. I have some major concerns need to be addressed before the paper could be accepted.

Q1: The introduction of previous works about channel selection methods using fNIRS-BCI are not enough, and the authors are suggested to add more recent works about channel selection methods using fNIRS-BCI. Some highly related papers on channel selection methods using fNIRS-BCI also should be included and commented in the literature review, including:

[1] Bispectrum-Based Channel Selection for Motor Imagery Based Brain-Computer Interfacing. IEEE Transactions on Neural Systems and Rehabilitation Engineering, 2020.

[2] EEG-based Emotion Recognition via Channel-wise Attention and Self Attention, IEEE Transactions on Affective Computing, 2020.

Q2: How is the parameter setting of the proposed method and the compared methods? Have they tuned to the best performance?

Q3: More channel selection compared methods should be added to demonstrate the efficiency of the proposed method.

Q4: More classification methods should be added to demonstrate the efficiency of different channel selection methods.

Q5: Why the classification accuracy of the t-value method is lower than that of all channel method?

Q6: It would be better to conduct experiments with more datasets.

Q7: The linguistic quality of this paper should be improved.

Author Response

This paper proposed a channel selection method based on z-score to improve fNIRS-BCI performance. On the whole, the part of introduction is not enough, and more experiments should be added to demonstrate the advantage of the proposed method. I have some major concerns need to be addressed before the paper could be accepted.

Response: Authors are really grateful to the reviewer for thorough review of the manuscript and providing the valuable comments to improve the quality of this paper. The manuscript has been thoroughly revised upon the comments and suggestions of the reviewer.

 Comment 1: The introduction of previous works about channel selection methods using fNIRS-BCI are not enough, and the authors are suggested to add more recent works about channel selection methods using fNIRS-BCI. Some highly related papers on channel selection methods using fNIRS-BCI also should be included and commented in the literature review, including:

[1] Bispectrum-Based Channel Selection for Motor Imagery Based Brain-Computer Interfacing. IEEE Transactions on Neural Systems and Rehabilitation Engineering, 2020.

[2] EEG-based Emotion Recognition via Channel-wise Attention and Self Attention, IEEE Transactions on Affective Computing, 2020.

Response: The literature review includes previous works on fNIRS-BCI primarily. The EEG-BCI based works were partially included to prove the significance of channel selection methods. However, as per reviewer’s suggestion introduction section is updated with more significant, latest and related literature review including the mentioned articles. Thank you for highlighting the mentioned papers.

 Comment 2: How is the parameter setting of the proposed method and the compared methods? Have they tuned to the best performance?

Response:  The tuning of parameters is not performed for t-value and z-score method. Fixed values of parameters involved in calculation of cHRF are used as per previous referred studies and NIRS-SPM toolbox. Two-gamma function is used to calculate cHRF.

 Comment 3: More channel selection compared methods should be added to demonstrate the efficiency of the proposed method.

Response: For fNIRS-BCI studies, t-value method has been used excessively as channel selection method. Few previous studies have used baseline correction technique, which selects channel if the peak value of task is greater than the peak value of rest for recorded brain signal. In this paper,  t-value method is used as benchmark method because it is found in most of recent studies. Other than t-value method no proper method has been devised to select channels for fNIRS-BCI, however channel selection methods may exist for EEG-BCIs and fMRI-BCIs. Following the reviewers comment, baseline correction technique is applied and results have been discussed in the revised manuscript. Thank you!

Comment 4: More classification methods should be added to demonstrate the efficiency of different channel selection methods.

Response: This is a very good suggestion. In the current work LDA was used. LDA is a commonly used classifier for the fNIRS-BCI systems; 45% fNIRS-BCI studies applied LDA as a classification method in their studies [49]. It was selected as classification method because of its low computational cost and high-speed performance. Since the major aim of this study was the comparison of different channel selection methods, only LDA has been used. In future works, more classifiers can be explored. Thank you for this suggestion.

Comment 5: Why the classification accuracy of the t-value method is lower than that of all channel method?

Response: Following the literature, same steps and procedure are implemented to calculate t-values to select cortically active channel. The method select channels on the basis of statistical significance of channels data i.e., if t > tcrt and p-value < 0.05. Cortical activation is measured as statistical significance to select channel using t-value method. Also figure 8 shows that number of channel selected using t-value method ranges from 6-36 for MA, 2-24 for RMI and 6-36 for LMI tasks. The variation in selecting total number of channels is the main reason this method yielded lower classification accuracies with respect to z-score method and all channels data. 

Comment 6: It would be better to conduct experiments with more datasets.

 Response: As per reviewer’s suggestion, complete analysis has been performed on open access database containing fNIRS brain signals of 17 subjects for left-hand finger tapping, right-hand finger tapping and dominant foot tapping tasks. The database contains 25 trials of each activity for each subject. The results also validated the high performance of proposed method over conventional t-value method and baseline correction technique. Thank you for the valuable suggestion.

Comment 7: The linguistic quality of this paper should be improved.

Response: With reference to your kind comment, the manuscript has been thoroughly revised to improve the linguistic quality, including tenses, academic writing style and sentence structure.

Reviewer 2 Report

The authors present a new method for performing channel selection in BCIs based on fNIRS technology. Though the results indicate good performance of the method a number of methodological flaws exist in the manuscript and it also lacks proper comparison with the state of the art. Some of the results are unexpected and are not discussed in the manuscript. Finally, the contribution of the method is marginal and unless authors explore the reasons of improved performance over state of the art in more detail and validate it on another data set, it is more appropriate for a conference publication than a journal manuscript.

General remarks:

  • There are a number of incorrect/misleading statements: fNIRS is not measuring brain activity but blood oxygenation, fNIRS does not have better temporal resolution than fMRI neither better spatial resolution compared to (high-density) EEG, fNIRS are not that user friendly compared to (low-channel) EEG, etc.
  • Although identifies as a separate step of a BCI pre-processing step, channel selection is often done as part of other steps in the signal processing / classification pipeline (e.g., feature selection), hence can be approached using different methods, as also mentioned in some of the state of the art
  • There is no details on which validation scheme was applied – there is no indication of splitting the date in train-test(-validation) set.
  • The classification results are suspicious. They indicate that using all channels gives better results than using the selected ones after the application of t-value method. This suggest of a potential error in the t-value method application (also suggested by the graphs representing feature distribution)
  • Although authors suggest that they demonstrate improvements over state of the art – they only compared their method to a t-value channel selection method. A multitude of methods exist that report better performance, some of them being cited in the manuscript

More detailed remarks:

  • The abstract include abbreviations that are not introduced: fNIRS, LMI, and RMI.
  • Figure 4 is superficial as a and b part are the same except the method used for channel selection;
  • There is an issue in equation 2
  • There is a mislabelling of figure 5. It has never been referred to in text though I assume early Fig 6 references are pointing to this one. Figure 7-9 should refer to 6-8.
  • Figure 5 (Fig 6 in text) does not give an information of which channels are selected. Also, it is difficult to compare the t-value to z-score methods due to overall differences in the value ranges and distributions
  • Tables 1-3 are too detailed and not that informative. A graph with statistics on total number of (common) channels would be more informative
  • Giving statistically significant results in a table formatted as Table 5 is not a common practice
  • The discussion section is essentially a repetition of the introduction without a proper discussion of results and limitations are poorly described

Author Response

The authors present a new method for performing channel selection in BCIs based on fNIRS technology. Though the results indicate good performance of the method a number of methodological flaws exist in the manuscript and it also lacks proper comparison with the state of the art. Some of the results are unexpected and are not discussed in the manuscript. Finally, the contribution of the method is marginal and unless authors explore the reasons of improved performance over state of the art in more detail and validate it on another data set, it is more appropriate for a conference publication than a journal manuscript.

Response: Authors are really grateful to the reviewer for thorough reviewing the manuscript and providing the valuable comments to improve the quality of this paper. The proposed method has also been validated on another open access database and yield better performance over conventional t-value method. The manuscript has been thoroughly revised upon the comments and suggestions of the referee.

General remarks:

Comment 1: There are a number of incorrect/misleading statements: fNIRS is not measuring brain activity but blood oxygenation, fNIRS does not have better temporal resolution than fMRI neither better spatial resolution compared to (high-density) EEG, fNIRS are not that user friendly compared to (low-channel) EEG, etc.

Response: Reviewer is right! fNIRS records blood oxygenation level changes for cortical brain activation. fNIRS has low spatial resolution as compare to fMRI and high-density EEG, but the advantage of low cost, portability and wear-ability makes it a favorable choice for research.  The concerned sentence has been removed from the revised manuscript. Thank you!

Comment 2: Although identifies as a separate step of a BCI pre-processing step, channel selection is often done as part of other steps in the signal processing / classification pipeline (e.g., feature selection), hence can be approached using different methods, as also mentioned in some of the state of the art

Response: Reviewer is right! The channel selection is performed after noise removal step. As per reviewers comment, the channel selection is now mentioned as first step of feature extraction in the figure and text. Thank you for the correction.

Comment 3: There is no details on which validation scheme was applied – there is no indication of splitting the data in train-test (-validation) set.

Response: Leave-one-out cross-validation scheme is applied and also mentioned in heading 2.8. The dataset is divided into training and testing sets, to ensure separation of data for training and testing of classifier for each channel selection method and activity used. This sentence has also been added in heading 2.8 in revised manuscript. Thank you for highlighting the missing information.

Comment 4: The classification results are suspicious. They indicate that using all channels gives better results than using the selected ones after the application of t-value method. This suggest of a potential error in the t-value method application (also suggested by the graphs representing feature distribution)

Response: Following the literature, same steps and procedure are implemented to calculate t-values to select cortically active channel. The t-value method was first used to measure the statistical significance of channel. The method select channels on the basis of statistical significance of channels data i.e., if t > tcrt and p < 0.05. Also figure 8 shows that number of channel selected using t-value method ranges from 6-36 for MA, 2-24 for RMI and 6-36 for LMI tasks. The variation in selecting total number of channels is the main reason this method yielded lower classification accuracies with respect to z-score method and all channels data.

Comment 5: Although authors suggest that they demonstrate improvements over state of the art – they only compared their method to a t-value channel selection method. A multitude of methods exist that report better performance, some of them being cited in the manuscript

Response: For fNIRS-BCI studies, t-value method has been used excessively as channel selection method. Few previous studies have used baseline correction technique, which selects channel if the peak value of task is greater than the peak value of rest for recorded brain signal. In this paper,  t-value method is used as benchmark method because it is found in most of recent studies. Other than t-value method no proper method has been devised to select channels for fNIRS-BCI, however channel selection methods may exist for EEG-BCIs and fMRI-BCIs. Following the reviewers comment, baseline correction technique is applied and results have been discussed in the revised manuscript. Thank you!

More detailed remarks:

Comment 6: The abstract include abbreviations that are not introduced: fNIRS, LMI, and RMI.

Response: Yes reviewer is right, these abbreviations has now been introduced in abstract section. Thank you for the correction.

Comment 7: Figure 4 is superficial as a and b part are the same except the method used for channel selection;

Response: Yes the reviewer is right, this figure was split into (a) and (b) part just to highlight the change in channel selection method as conventional and proposed methodology. This may help the reader to understand the basic difference and highlights the contribution of this study.

Comment 8: There is an issue in equation 2

Response: Thank you for highlighting the mistake. The equation has been corrected in the revised manuscript. 

Comment 9: There is a mislabelling of figure 5. It has never been referred to in text though I assume early Fig 6 references are pointing to this one. Figure 7-9 should refer to 6-8.

Response: The figure numbers has been updated and corrected in the revised manuscript. All figures are cited in text. Thank you for the correction.

Comment 10: Figure 5 (Fig 6 in text) does not give an information of which channels are selected. Also, it is difficult to compare the t-value to z-score methods due to overall differences in the value ranges and distributions

Response: Yes, the figure does not show the information of selected channels; it shows the difference in measuring cortical activation by using both methods. The t-value method measures the activation as statistical significance of channel, while z-score measures the activation as z-score of (max of) cross-correlation coefficient with respect to all channels population. The activation map is drawn for both methods using normalized t-values and z-scores. Figure shows the cortical activation-map of the t-value and z-score method for LMI, RMI, and MA tasks. The figure number has been updated and corrected in the revised manuscript. Thank you for correcting the figure number.

 Comment 11: Tables 1-3 are too detailed and not that informative. A graph with statistics on total number of (common) channels would be more informative

Response: As per reviewer suggestion, these tables are now available as supplementary files. Figure 8 has been added, showing total number of channels using t-value and z-score method along-with common channels for MA, RMI and LMI tasks. Thank you for the valuable suggestion.

Comment 12: Giving statistically significant results in a table formatted as Table 5 is not a common practice

Response: In order to ease the understanding and readability of the reader, complete results of statistically significance test has been shown in Table 5 (now Table 2 in revised manuscript) for all three activities.

Comment 13: The discussion section is essentially a repetition of the introduction without a proper discussion of results and limitations are poorly described

Response: As per reviewers comment, discussion section has been revised. First previous studies are discussed related to channel selection methods of fNIRS-BCI. Then proposed method and its experimental results are discussed with improvement in performance. Followed by reasoning and detailed comparison with t-value method, and then limitations are revised. Thank you for the suggestion.

Reviewer 3 Report

The manuscript proposes a new method (the z-score method) to select informative channels for classifying fNIRS data. An open-source dataset of rest, Left Motor Imagery, Right Motor Imagery and Mental Arithmetic is used. The study performs LDA classification for classifying rest vs. LMI, rest vs. RMI and rest vs. MA and compares accuracies between the t-value channel selection method, the z-score channel selection method and using all channels. The authors find a significant increase in classification accuracy for all tasks when using the z-score method as compared to the other two methods. The study is interesting and I believe channel selection is an ongoing challenge for most BCI applications. However, I have some concerns that must be addressed in order to properly be able to assess the work. These concerns are expressed through the following comments.

  1. It is not clear whether train and train sets were set up in order to keep the data used in the channel selection separate from the data used in the LDA testing (using LOO). If data sets were not separated, it would mean that the classification accuracies are inflated, and the same accuracies would with high probability not be obtainable in a real-time scenario. Also comparing all channels and z-score/t-value methods would in that case not be fair.

  1. In equation 2, first what is meant by “in each stimulus”? Does it mean for each mental task? Second, if ‘k’ is the stimulus type, it is correct that k can only take 2 values? Either 0 for rest or 1 for task (LMI, RMI or MA), is that correct? If that is the case, equation 4 is a bit unclear to me (see next comment).

  1. In equation 4, shouldn’t the sum go up to N-1? What does k denote here? And also in s(k-n)? This equation is not clear, please clarify.

  1. The rationale behind the cross-correlation as expressed in equation 7 is not entirely clear to me. From my understanding rxy would be a vector of cross-correlation coefficients signifying the similarity between the measured hemodynamic response and time-shifted versions (by changing tau) of the desired hemodynamic response. What I don’t understand is why you would want to just choose the highest coefficient without considering tau? Please clarify this aspect as I believe it would also strengthen impact of the manuscript.

  1. Then in eq 5, it is not clear how the z-score can be below 0 for any cross-correlation coef. that is positive (within the r-vector). Since zi represents the max and the mean is subtracted from the max, the z-score will always be positive except maybe for negative cross-correlations. I believe it would be appropriate to clarify by showing the distribution of cross-correlation coefficients, to get a better understanding of the typical behavior (goes to comment 4).

  1. Judging from figure 6-8, it seems like it’s only the peak-value that allows to discriminate between rest and task (LIM, RIM and MA). What was the reason to also include mean and slope?

  1. In figure 2, what are the blue circles?

  1. Page 10: Should be Figures 6, 7, and 8.

  1. Figure 10 should be figure 9.

  1. A discussion about the only partial overlap of channels between the t-value and z-score methods would be very useful for the reader. The methods did seem to pick very different channels, why? The channels picked for subject 28 do not seem to be entirely what one would expect during LMI and RMI, why? To better map out the discrepancy between the two methods and try to discuss why the differ and how the selected channels reflect expected brain patterns is very important for the scientific community and would increase the impact of the study.

  1. Last, please do a thorough re-read of the article as some phrasings are a bit awkward, especially in the Materials and Methods section. For example: page 4, line 144.

Author Response

The manuscript proposes a new method (the z-score method) to select informative channels for classifying fNIRS data. An open-source dataset of rest, Left Motor Imagery, Right Motor Imagery and Mental Arithmetic is used. The study performs LDA classification for classifying rest vs. LMI, rest vs. RMI and rest vs. MA and compares accuracies between the t-value channel selection method, the z-score channel selection method and using all channels. The authors find a significant increase in classification accuracy for all tasks when using the z-score method as compared to the other two methods. The study is interesting and I believe channel selection is an ongoing challenge for most BCI applications. However, I have some concerns that must be addressed in order to properly be able to assess the work. These concerns are expressed through the following comments.

Response: Authors are really grateful to the reviewer for thorough reviewing the manuscript and providing the valuable comments to improve the quality of this paper. The manuscript has been thoroughly revised upon the comments and suggestions of the referee.

Comment 1: It is not clear whether train and train sets were set up in order to keep the data used in the channel selection separate from the data used in the LDA testing (using LOO). If data sets were not separated, it would mean that the classification accuracies are inflated, and the same accuracies would with high probability not be obtainable in a real-time scenario. Also comparing all channels and z-score/t-value methods would in that case not be fair.

Response: Leave-one-out cross-validation scheme is applied and also mentioned in heading 2.8. The dataset is divided into training and testing sets, to ensure separation of data for training and testing of classifier for each channel selection method and activity used. This sentence has also been added in heading 2.8 in revised manuscript. Thank you for highlighting the missing information.

Comment 2: In equation 2, first what is meant by “in each stimulus”? Does it mean for each mental task? Second, if ‘k’ is the stimulus type, it is correct that k can only take 2 values? Either 0 for rest or 1 for task (LMI, RMI or MA), is that correct? If that is the case, equation 4 is a bit unclear to me (see next comment).

Response: In equation 2 there was a typo mistake, ‘k’ represents the discrete time values of the brain signal. The typo mistake has been corrected for equation 2 and 4 in the revised manuscript. Thank you for the correction.

Comment 3: In equation 4, shouldn’t the sum go up to N-1? What does k denote here? And also in s(k-n)? This equation is not clear, please clarify.

Response: Two variables ‘k’ and ‘N’ were wrongly used for the same notation in the previous manuscript. All equations has been corrected with the single variable ‘k’ to represent discrete time in the revised manuscript. In equation 4,  is the desired hemodynamic response (dHRF) calculated by convolving modeled hemodynamic response function (cHRF) with boxcar function s(k). The s(k-n) is the reversed and shifted boxcar function s(k).  Thank you for the correction.

Comment 4: The rationale behind the cross-correlation as expressed in equation 7 is not entirely clear to me. From my understanding rxy would be a vector of cross-correlation coefficients signifying the similarity between the measured hemodynamic response and time-shifted versions (by changing tau) of the desired hemodynamic response. What I don’t understand is why you would want to just choose the highest coefficient without considering tau? Please clarify this aspect as I believe it would also strengthen impact of the manuscript.

Response: The desired and measured hemodynamic response contains activity and rest period. Yes the rxy would be a vector of cross-correlation coefficients signifying the similarity between the measured hemodynamic response and time-shifted versions (by changing tau) of the desired hemodynamic response. Maximum strength of similarity occur when peak of desired and measured response overlap (task period of both responses), it is also the highest value of cross-correlation coefficient. Here if we choose tau for rest vs rest it will give lowest value of cross-correlation coefficient. And if don’t select any tau, calculated cross-correlation coefficient for complete time period, the highest value remains the same. Therefore selecting tau does not affect the occurrence of highest value. It has been added in the revised manuscript in heading 2.6.2. Thank you for the valuable comment.

Comment 5: Then in eq 5, it is not clear how the z-score can be below 0 for any cross-correlation coef. that is positive (within the r-vector). Since zrepresents the max and the mean is subtracted from the max, the z-score will always be positive except maybe for negative cross-correlations. I believe it would be appropriate to clarify by showing the distribution of cross-correlation coefficients, to get a better understanding of the typical behavior (goes to comment 24).

Response: The vector  contains maximum values of cross-correlation coefficients for each channel. The magnitude of each maximum value varies for each channel and form a new range. Therefore, forming a new mean value for vector ,  hence z-score can be less than zero. As per reviewer suggestion figure 5 (new) has been added containing plot of cross-correlation coefficients of all channels for LMI, plot of vector  containing maximum value of all channels for LMI and plot of z-score for LMI, RMI and MA task of subject 28. Thank you for the wonderful suggestion.

Comment 6: Judging from figure 6-8, it seems like it’s only the peak-value that allows to discriminate between rest and task (LIM, RIM and MA). What was the reason to also include mean and slope?

Response: In literature, peak mean and slope feature-combination was found to be optimal feature-combination yielding maximum classification accuracy for fNIRS-BCI [51]. Following the literature this feature combination is used for classification.

Comment 7: In figure 2, what are the blue circles?

Response: The Figure 2 represents optodes placement configuration for data recording. This figure is re-produced following the original image for open access database. The EEG and NIRS data was recorded simultaneously. For EEG electrodes are represented by blue and black (ground) circles and for NIRS red and green squares shows sources and detectors respectively.

Comment 8: Page 10: Should be Figures 6, 7, and 8.

Response: Figure numbers has been updated and corrected in the revised manuscript. Thank you for the correction.

Comment 9: Figure 10 should be figure 9.

Response: Figure numbers has been updated and corrected in the revised manuscript. Thank you for the correction.

Comment 10: A discussion about the only partial overlap of channels between the t-value and z-score methods would be very useful for the reader. The methods did seem to pick very different channels, why? The channels picked for subject 28 do not seem to be entirely what one would expect during LMI and RMI, why? To better map out the discrepancy between the two methods and try to discuss why the differ and how the selected channels reflect expected brain patterns is very important for the scientific community and would increase the impact of the study.

Response: The t-value method measures the cortical activation as statistical significance of channel’s data, while z-score measures the activation as z-score of (max of) cross-correlation coefficient of channel with respect to (max of) cross-correlation coefficient of all channels population. The activation map is drawn for both methods using normalized t-values and z-scores. It can be seen in Figure 7, cortical activation does not occur in all channels for a specific activity, and also not in all channels of the designated region. For MA task the proposed method selected major channels from prefrontal region (details can be found in supplementary files) and visual cortex (instructions were displayed on screen), however some other channels from motor cortex also showed positive z-score value. Similarly for LMI and RMI major channels are selected from motor & prefrontal cortex (details can be found in supplementary files) and visual cortex (instructions were displayed on screen. The areas of activation found are similar to designated areas of neural activity [54]. The selection of extra channels is may be due to human error, lack of concentration, multiple thinking during experiment and induced neuronal activity. This also varies with respect to activity and subject. This also has been added in result section. Thank you for the valuable comment.

Comment 11: Last, please do a thorough re-read of the article as some phrasings are a bit awkward, especially in the Materials and Methods section. For example: page 4, line 144.

Response: Thank you for highlighting the mistake, it has been corrected in the revised manuscript.

Reviewer 4 Report

Authors have reported a method for channel selection for fNIRS BCI.

Comments to improve the paper:

  1. The importance of using z-score is not clearly discussed.
  2. The justification of the implementing z-score with respect to the correlation of the fNIRS signal is not clear.
  3. The study has used the data from online database. But the study mentions that the participants were paid. Is the funding used  for this study? Is authors recorded the data? if not it is misrepresenting in this paper. 
  4. How well the Z-score change with respect to different task has to be discussed.
  5. What is the advantage of using canonical hemodynamic response function (cHRF) 2-gamma functions? How this is justified? Why it has been selected or just used from the referred study?
  6. Is figure 1 created by Authors ? if not please mention the permission information 

Author Response

Authors have reported a method for channel selection for fNIRS BCI.

Response: Authors are really grateful to the reviewer for thorough reviewing the manuscript and providing the valuable comments to improve the quality of this paper. The manuscript has been thoroughly revised upon the comments and suggestions of the referee.

Comments to improve the paper:

Comment 1: The importance of using z-score is not clearly discussed.

Response: z-score measures the distance of raw score from mean value i.e., how far from mean a data point is in population. In this study, z-score represents the channel activation in form of matching and strength of similarity based on the value of cross-correlation coefficient. Positive z-score represents higher strength & similarity and negative z-score shows lower strength & no matching. Thank you for the suggestion, it has been added in the revised manuscript in section 2.6.2

Comment 2: The justification of the implementing z-score with respect to the correlation of the fNIRS signal is not clear.

Response: Cross-correlation measures the strength of similarity between desired and measured hemodynamic response in form of cross-correlation coefficient. Higher the value of cross-correlation coefficient higher the strength of similarity. For each channel highest value of cross-correlation coefficient is selected and formed a vector of max values. Z-score of this vector is then calculated to measures the strength & similarity desired and measured hemodynamic responses.. Positive z-score represents higher strength & similarity and negative z-score shows lower strength & no matching. Thank you for the suggestion, it has been added in the revised manuscript in section 2.6.2

Comment 3: The study has used the data from online database. But the study mentions that the participants were paid. Is the funding used for this study? Is authors recorded the data? if not it is misrepresenting in this paper. 

Response: No, open access data base has been used in this study. It was a typo, the original authors of database recorded the data and utilized funding. This sentence regarding funding has been removed in the revised manuscript. Proper citation and acknowledgment has been made to regard the original authors of the database. Thank you for the correction.

Comment 4: How well the Z-score change with respect to different task has to be discussed.

Response: z-score value of any channel depends upon the maximum value of cross-correlation coefficient corresponding to that channel. The variation of cross-correlation coefficient value is based on the strength of similarity between desired and measured hemodynamic response for specific activity of that channel. The analysis shows that z-score varies with respect to activity (LMI, RMI and MA), as shown in Figure 5(c). Also z-score varies with subject.  Thank you for the suggestion.

Comment 5: What is the advantage of using canonical hemodynamic response function (cHRF) 2-gamma functions? How this is justified? Why it has been selected or just used from the referred study?

Response: Following the literature and previous studies, for estimation of canonical hemodynamic response function (cHRF) 2-gamma functions is used. 3-gamma function is used to estimate cHRF with initial-dip occurrence. In this study, initial-dip analysis has not been performed; therefore 2-gamma function is used to estimate cHRF.

Comment 6: Is figure 1 created by Authors ? if not please mention the permission information 

Response: Figure 1 is inspired by a previous figure for which the reference is provided. It was created by the authors, however. Thank you very much!

Round 2

Reviewer 1 Report

The revised manuscript has solved my concern, and I suggest accepting this paper.

Author Response

Reviewer 1:

The revised manuscript has solved my concern, and I suggest accepting this paper.

Response: Authors are really grateful to the reviewer for thorough review of the revised manuscript and providing the valuable comments to improve the quality of this paper. Thankyou!

Reviewer 2 Report

The authors have substantially improved the manuscript, taking most of the comments on board. However, there are few aspects that should be improved. One of the main one is to have a proof reading (native speaker) of the text and better discussion and conclusion sections. The conclusion section is missing context - summary statement of the manuscript contribution. Also, the new figures (Fig 5b and 5c and Fig 8) introduced are not in the right format - they display discrete data points (not time series data).

Comments on the response

Response: Authors are really grateful to the reviewer for thorough reviewing the manuscript and providing the valuable comments to improve the quality of this paper. The proposed method has also been validated on another open access database and yield better performance over conventional t-value method. The manuscript has been thoroughly revised upon the comments and suggestions of the referee.

- The authors should also introduce the main reasoning of including this validation in the intro of the paper as well as discussing it in the methods part.

Comment 10: Figure 5 (Fig 6 in text) does not give an information of which channels are selected. Also, it is difficult to compare the t-value to z-score methods due to overall differences in the value ranges and distributions

Response: Yes, the figure does not show the information of selected channels; it shows the difference in measuring cortical activation by using both methods. The t-value method measures the activation as statistical significance of channel, while z-score measures the activation as z-score of (max of) cross-correlation coefficient with respect to all channels population. The activation map is drawn for both methods using normalized t-values and z-scores. Figure shows the cortical activation-map of the t-value and z-score method for LMI, RMI, and MA tasks. The figure number has been updated and corrected in the revised manuscript. Thank you for correcting the figure number.

- This explains indeed what is represented in the figure, however it is not clear how is difference in cortical activation shown in the figure translated in channel selection and where do the differences in activation come from. Why there is spatial difference between the methods? This is just an illustration on one subject. Are the others showing the same pattern? 

Comment 11: Tables 1-3 are too detailed and not that informative. A graph with statistics on total number of (common) channels would be more informative

Response: As per reviewer suggestion, these tables are now available as supplementary files. Figure 8 has been added, showing total number of channels using t-value and z-score method along-with common channels for MA, RMI and LMI tasks. Thank you for the valuable suggestion.

- As figure 8 is representing discrete data points, different format should be used - point graph of bar chart

Comment 13: The discussion section is essentially a repetition of the introduction without a proper discussion of results and limitations are poorly described

Response: As per reviewers comment, discussion section has been revised. First previous studies are discussed related to channel selection methods of fNIRS-BCI. Then proposed method and its experimental results are discussed with improvement in performance. Followed by reasoning and detailed comparison with t-value method, and then limitations are revised. Thank you for the suggestion.

- Although the authors have slightly improved the discussion section, it can be further enhanced. The main aspects could be in the direction of application of such a method in broader context of fNIRS-BCI as well as limitations and potential improvements / adaptations.

Author Response

Reviewer 2:

The authors have substantially improved the manuscript, taking most of the comments on board. However, there are few aspects that should be improved. One of the main one is to have a proof reading (native speaker) of the text and better discussion and conclusion sections. The conclusion section is missing context - summary statement of the manuscript contribution. Also, the new figures (Fig 5b and 5c and Fig 8) introduced are not in the right format - they display discrete data points (not time series data).

Response: Authors are really grateful to the reviewer for thorough reviewing the revised manuscript and providing the valuable comments to further improve the quality of this paper. As per reviewer suggestion, the manuscript has been proof read. Discussion and conclusion sections have been revised, including the addition of context-summary statement in conclusion section.  Also Figure 5 and 8 have been re-drawn in the right format. The manuscript has been thoroughly revised upon the comments and suggestions of the reviewer.

Comments on the response

Response: Authors are really grateful to the reviewer for thorough reviewing the manuscript and providing the valuable comments to improve the quality of this paper. The proposed method has also been validated on another open access database and yield better performance over conventional t-value method. The manuscript has been thoroughly revised upon the comments and suggestions of the referee.

New Comment: The authors should also introduce the main reasoning of including this validation in the intro of the paper as well as discussing it in the methods part.

Response: As per reviewer comment, the reason of applying proposed method on another dataset has been discussed in the introduction section. Thankyou!

R1_Comment 10: Figure 5 (Fig 6 in text) does not give an information of which channels are selected. Also, it is difficult to compare the t-value to z-score methods due to overall differences in the value ranges and distributions

R1_Response: Yes, the figure does not show the information of selected channels; it shows the difference in measuring cortical activation by using both methods. The t-value method measures the activation as statistical significance of channel, while z-score measures the activation as z-score of (max of) cross-correlation coefficient with respect to all channels population. The activation map is drawn for both methods using normalized t-values and z-scores. Figure shows the cortical activation-map of the t-value and z-score method for LMI, RMI, and MA tasks. The figure number has been updated and corrected in the revised manuscript. Thank you for correcting the figure number.

New Comment: This explains indeed what is represented in the figure, however it is not clear how is difference in cortical activation shown in the figure translated in channel selection and where do the differences in activation come from. Why there is spatial difference between the methods? This is just an illustration on one subject. Are the others showing the same pattern?

Response: The spatial difference of identifying cortical activation is found in both methods because of the fact that both methods applies different scientific and mathematical principles; t-value method uses statistical significance by GLM and z-score method applies signal matching through cross-correlation. Similar pattern of differences are found in all subjects. This explanation has also been added in the results section. Thank you for the valuable comment.

R1_Comment 11: Tables 1-3 are too detailed and not that informative. A graph with statistics on total number of (common) channels would be more informative

R1_Response: As per reviewer suggestion, these tables are now available as supplementary files. Figure 8 has been added, showing total number of channels using t-value and z-score method along-with common channels for MA, RMI and LMI tasks. Thank you for the valuable suggestion.

New Comment: As figure 8 is representing discrete data points, different format should be used - point graph of bar chart

Response: Figure 8 has been re-drawn as discrete dot plot in the revised manuscript. Thank you for your valuable suggestion.

R1_Comment 13: The discussion section is essentially a repetition of the introduction without a proper discussion of results and limitations are poorly described

R1_Response: As per reviewers comment, discussion section has been revised. First previous studies are discussed related to channel selection methods of fNIRS -BCI. Then proposed method and its experimental results are discussed with improvement in performance. Followed by reasoning and detailed comparison with t-value method, and then limitations are revised. Thank you for the suggestion.

New Comment: Although the authors have slightly improved the discussion section, it can be further enhanced. The main aspects could be in the direction of application of such a method in broader context of fNIRS-BCI as well as limitations and potential improvements / adaptations.

Response: The z-score method is discussed in context of fNIRS-BCI applications and uses. Limitations of the study has also been revised in the discussion section. Thank you for the valuable comment.

Reviewer 4 Report

Authors have satisfactorily addressed the comments..

minor issues:

  1. results in numbers should not be in the introduction..it seems repetitive from abstract.
  2. please mention the number of training and testing data atleast in percentages

Author Response

Reviewer 4:

Authors have satisfactorily addressed the comments.

Response: Authors are really grateful to the reviewer for thorough reviewing the revised manuscript and providing the valuable comments to improve the quality of this paper. The manuscript has been thoroughly revised upon the comments and suggestions of the referee.

Minor issues:

Comment 1: Results in numbers should not be in the introduction. It seems repetitive from abstract.

Response: As per reviewer comment, the results in numbers have been removed from introduction section. Thankyou!

Comment 2: Please mention the number of training and testing data at least in percentages

Response: Because of limited number of samples i.e., twenty, LOOCV is applied. One sample for testing and nineteen for training the classifier, repeated twenty times. The information has been added in the revised manuscript in section 2.8.